# Association between Vitamin D and Cognitive Deficiency in Alcohol Dependence

**DOI:** 10.3390/healthcare10091772

**Published:** 2022-09-14

**Authors:** Visnja Banjac Baljak, Goran Mihajlovic, Nera Zivlak-Radulovic, Lana Nezic, Mirjana Miskovic, Vesna Banjac

**Affiliations:** 1Clinic of Psychiatry, University Clinical Center of the Republic of Srpska, 78 000 Banjaluka, Bosnia and Herzegovina; 2Department of Psychiatry, Faculty of Medical Sciences, University of Kragujevac, 34 000 Kragujevac, Serbia; 3Department of Pharmacology, Toxicology and Clinical Pharmacology, Faculty of Medicine, University of Banja Luka, 78 000 Banja Luka, Bosnia and Herzegovina; 4Department of Pharmacy, Faculty of Medicine, University of Banja Luka, 78 000 Banja Luka, Bosnia and Herzegovina

**Keywords:** alcohol dependence, cognitive deficiency, vitamin D deficiency, Montreal Cognitive Assessment, Addenbrooke’s Cognitive Examination-Revised

## Abstract

There are still not enough findings to elucidate how exactly alcohol use impairs cognitive abilities. Some studies have shown that there is a link between alcohol intake and vitamin D levels, but these findings are inconsistent so further research is needed. The aim of this study was to investigate the association between serum vitamin D levels and cognitive impairment in alcohol-dependent individuals. A case-control study was carried out including a total of *N* = 132 respondents with a medical history of alcoholism, and healthy volunteers. The Montreal Cognitive Assessment (MoCa) and Addenbrooke’s Cognitive Examination-Revised (ACE-R) screening tools were used for cognitive status assessment and serum vitamin D levels analysis (blood samples of respondents). Significant difference (*p* = 0.022), was found in vitamin D levels in the alcohol-dependent group with cognitive deficiency 13.7 ± 9.4 (ng/mL), alcohol-dependent group without cognitive deficiency 19.5 ± 11.2 (ng/mL) and healthy controls 19.9 ± 11.1 (ng/mL), respectively. Furthermore, vitamin D levels were significantly different across all groups based on MoCa (*p* = 0.016) and ACE-R (*p* = 0.004) scores. All three groups exhibited vitamin D deficiency. A significant correlation was found between vitamin D deficiency and cognitive impairment, but it yielded no significant difference in alcohol-dependent individuals.

## 1. Introduction

Alcohol dependence is a chronic and recurring condition. It is often characterized by the desire to continue using alcohol despite the harmful effects it has on both physical and mental health. Other alcohol use-associated factors include loss of control over the amount or timing of alcohol use, tolerance, psychological and physical dependence, and abstinence syndrome [1]. Investigating brain structure in alcohol-dependent individuals has shown that excessive alcohol use leads to microglial activation, the release of proinflammatory mediators, and the induction of neuroinflammation. It also impairs neurogenesis, which altogether may interfere with neurotransmitter hyperactivity (glutamate), neuron loss, and cognitive impairment [2,3]. Alcohol use, misuse, and addiction are all associated with cognitive impairment. The prevalence of cognitive impairment has been documented in 50–80% of alcohol dependence [4]. The most affected cognitive functions are the ability to learn new skills, visuospatial ability, executive function and attention, while verbal abilities usually remain intact [5].

Vitamin D is a group of lipophilic hormones with pleiotropic actions. It has been traditionally related to bone metabolism, although several studies in the last decade have suggested its role in cardiovascular diseases, diabetes, malignancies, autoimmune diseases, and infections [6]. Vitamin D is responsible for calcium and phosphorus homeostasis, cell proliferation, differentiation and apoptosis, immune and hormonal regulation, and other processes in the body [7]. Vitamin D receptors (VDRs) are present in the central nervous system and neurons have the ability to synthesize 1,25-dihydroxyvitamin D, the active form of vitamin D. There have been a number of studies conducted to determine the correlation between mental disorders (depression, schizophrenia, autism, Alzheimer’s disease, other cognitive impairments, alcohol dependence) and vitamin D levels. However, these findings were inconsistent, which highlights the need for further research [8,9,10,11]. Evidence on the association between alcohol dependence and vitamin D is a subject of controversial scientific discussions. While some studies suggest that vitamin D deficiency in alcohol-dependent individuals is a result of reduced light exposure, poor nutrition, alcohol-induced liver damage, and malabsorption [12], others, which have found a positive correlation between alcohol use and vitamin D, indicate that little is known about how this biochemical mechanism really works [13]. In recent years, many experimental and clinical studies have shown that vitamin D deficiency increases the risk of cognitive impairment by 2.4 times [9,14,15,16]. Nevertheless, the true nature of the relationship between vitamin D and alcohol-induced disorders is still poorly understood [17].

The aim of this study was to investigate the association between serum vitamin D levels and the level of cognitive impairment in alcohol-dependent individuals and to assess indirectly the role of vitamin D in the development of this cognitive deficit.

## 2. Materials and Methods

### 2.1. Subjects

This case-control study was carried out between October 2019 and March 2020 at the Psychiatric Clinic and the Institute for Clinical Laboratory Diagnostics, the University Clinical Center of the Republic of Srpska (UCC RS), and the Institute for Transfusion Medicine of the Republic of Srpska after approval by the Ethics Committee to the UCC RS (01-9-371-2/18).

A total of *N* = 132 respondents were recruited in this study, alcohol-dependent individuals who had abstained from alcohol use for at least a month (33 alcohol-dependent individuals with cognitive deficiency and 33 alcohol-dependent individuals without cognitive deficiency) and 66 healthy controls. First, prior to any study procedure a written informed consent form was obtained, in accordance with the Good Clinical Practice (GCP) and Declaration of Helsinki and then, alcohol-dependent individuals were divided into two subgroups, those without cognitive deficiency (Alc) and those with cognitive deficiency (Alc + CD). The classification was based on psychiatric examination and screening tools Montreal Cognitive Assessment—MoCa (cut-off score was 26) [18] and Addenbrooke’s Cognitive Examination Revised—ACE-R (cut-off score was 82) [19]. The inclusion criteria were as follows: respondents with a clinical diagnosis of alcohol dependence (diagnostic criteria for ICD-10 classification), who had been treated and abstained from alcohol use for at least a month and had used alcohol for at least 5 years. The exclusion criteria included the following: a recent craniocerebral trauma, neurodegenerative diseases, cerebrovascular diseases, malignant diseases of all systems and organs, acute and chronic infectious, inflammatory and autoimmune diseases, heart failure, chronic diseases of the respiratory system, acute or chronic hepatic diseases, acute or chronic renal diseases, misuse of psychoactive substances, schizophrenia spectrum and other psychotic disorders, mood disorders, patients with dementia or other serious organic brain diseases, diseases of the endocrine system, blood diseases, osteoporosis and osteomalacia, use of vitamin D and calcium supplements, drugs that affect the metabolism of vitamin D and calcium, adherence to special dietary regimens, starvation and anorexia, use of doping agents or anabolic drugs, use of hormone therapy, pregnancy and lactation, as well as self-initiative withdrawal of respondents from further participation in the study. The control group consisted of healthy volunteers registered at the Institute for Transfusion Medicine, as blood donors. They also had to meet inclusion and exclusion criteria and undergo psychiatric examination and assessment based on screening tools MoCa and ACE-R. By meeting the inclusion and exclusion criteria, respondents were included consecutively from day one until reaching the total number, in accordance with the sample size.

### 2.2. Methods

Respondents were recruited according to their medical history and predefined research parameters: general information, personal and family history, and sociodemographic characteristics. MoCa and ACE-R scoring was used to assess cognition. The tests were administered immediately after enrollment. MoCa is a screening tool for rapid cognitive assessment. The Serbian version of the Montreal Cognition Assessment 7.1 test was used in this study. MoCa is a more recent test, designed in 1996 as a clinical tool for the detection of mild cognitive impairment. It assesses attention and concentration, executive function, memory, language, visuoconstructive ability, conceptualization, computation, and orientation. The first test is an alternative matching of numbers and letters (five each), then copying a cube and drawing a clock. The next two are visuospatial ability and executive function, followed by the animal naming test. In the short-term memory test, there are two attempts to repeat five common two-syllable words. The attention test requires repeating a series of numbers forward and then a different series backward. There is also vigilance and serial-7 subtraction test, too. Language abilities are tested through sentence repeating and phonetic fluency. In the abstract thinking test, one looks for similarities between pairs such as banana-orange, train-bike, and clock-ruler. This is followed by the delayed recall and short-term memory test described above. MoCa ends with the orientation test (date, month, year, day, place, and city). Scores on the MoCA range from zero to 30, with a score of 26 and higher generally considered normal. Respondents who have less than 12 years of formal education (elementary, high, or vocational school) are given a total of 1 point. A total score below 26 points indicates cognitive deficiency.

The ACE-R is a widely used test of cognitive function and it has been reported to have high specificity and sensitivity. The original version of the test was administered in Serbian. Necessary test adjustments had to be made because of the cultural characteristics of the study population. The Mini-mental State Examination (MMSE) is embedded in ACE-R. It covers five cognitive domains: orientation, registration, attention and calculation, recall and language (naming, repetition, 3-stage command, reading, writing, copying). ACE-R evaluates these domains: verbal fluency, memory, language and visuospatial processing. The achievement is assessed across 5 subscores: attention and orientation (18 points), short and long-term memory and recognition (26 points), verbal fluency (14 points), language (26 points), and visual and spatial skills (16 points). The maximum possible score is 100. Anything below 82 points indicates cognitive deficiency.

Blood samples were taken for analysis to determine standard laboratory and study parameters. The laboratory parameters were: blood sedimentation, complete blood count, Aspartate aminotransferase (AST), Alanine aminotransferase (ALT), Gamma glutamyltransferase (GGT), potassium, sodium, calcium, phosphorus, magnesium, urea and creatinine. The study parameter included screening for serum vitamin D level and blood sampling was done with respondents’ consent. Namely, a single fasting morning venous blood sample of 20 mL was obtained from each respondent between 7:00 and 9:00 a.m. Screening for serum vitamin D levels was evaluated by electrochemiluminescence immunoassay (ECLIA) on the Elecsys 2010 analyzer (Roche Diagnostics, Mannheim, Germany). Vitamin D total assay was used for the quantitative determination of total 25-OH vitamin D in human serum. The analytical measurement range of detection was 3.00–70.0 ng/mL, whereas the intra-assay coefficients of variation (CVs) were ≤5.5%, and the inter-assay CVs were ≤7.0. A reference range was 6.9–49.9 ng/mL with recommendation that desirable range should be above 30 ng/mL. Biochemical analyzes were conducted by a specialist in medical biochemistry. Although there is considerable discussion on the serum concentrations of 25-OH D associated with deficiency, the screening levels were matched against the commonly agreed ones. Accordingly, sufficient concentration vitamin D levels are ≥30 ng/mL (or ≥75 nmol/L). Vitamin D deficiency serum 25-OH D concentrations are <20 ng/mL (below 50 nmol/L), while vitamin D insufficiency serum 25-OH D concentrations are 21–29 ng/mL (50 to 75 nmol/L) [20].

### 2.3. Statistical Analysis

Statistical analysis was performed using analytic and descriptive statistics. Following data distribution using Kolmogorov-Smirnov and Shapiro–Wilk test, adequate reporting was accomplished through the arithmetic mean and standard deviation and value range or through the median and interquartile range with parametric test (One-way analysis variance ANOVA). The influence of individual independent variables was determined using the correlation coefficient (Spearman’s rank correlation). Pearson chi-square was used to present categorical variables with absolute numbers and percentages. They helped determine possible differences between the observed groups. The binary logistic regression analysis was used for identification of predictive variables. There was statistical significance on all scores with *p* < 0.05 or 95% confidence interval. The analysis was conducted using the IBM Statistics SPSS v 23.0 statistical package.

## 3. Results

### 3.1. Sample Characteristics

Table 1 shows the sociodemographic characteristics of respondents. Statistical analysis indicates that there was significant difference between three groups by the following data: age (*p* = 0.023); marital status (*p* = 0.014); education (*p* = 0.000); employment (*p* = 0.000) and financial status (*p* = 0.000).

Table 2 shows the laboratory parameters of respondents. Statistical analysis indicates that there was a significant difference between three groups by the following data: red blood counts (*p* = 0.000), white cell counts (*p* = 0.016), hemoglobin (*p* = 0.000), hematocrit (*p* = 0.000), platelet count (*p* = 0.001), creatinine (*p* = 0.000), potassium (*p* = 0.042), magnesium (*p* = 0.000), AST (*p* = 0.000), ALT (*p* = 0.000), GGT (*p* = 0.000).

### 3.2. Association between Vitamin D Status and Cognition

Of 132 respondents, 115 exhibited low vitamin D levels (85 with deficiency and 30 with insufficiency). Respondents showed low vitamin D levels, deficiency, and insufficiency as follows: in the Alc + CD group 25 and 5, respectively, in the Alc group 22 and 5 respectively, and in controls 39 and 20, respectively.

The mean of serum vitamin D in the Alc + CD group was 13.7 ± 9.4 (ng/mL), in the Alc 19.5 ± 11.2 (ng/mL) and in controls 19.9 ± 11.1 (ng/mL) (*F* (2129) = 3.940; *p* = 0.022), respectively (Figure 1).

The mean MoCa score was 23.5 ± 1.5 in the Alc+CD group, 27.2 ± 1.3 in the Alc, and 27.9 ± 1.1 in controls, respectively. There was statistical difference (F (2129) = 133.688; *p* < 0.0001). The mean ACE-R score was 74.24 ± 6.1 in the Alc + CD group, 87.9 ± 4.2 in the Alc, and 92.9 ± 3.9 in controls, respectively. There was a statistical difference (F (2129) =184.575; *p* < 0.0001).

Figure 2a,b show the association between serum vitamin D levels and cognitive function across all three groups. Vitamin D values were matched against MoCa (ϱ = 0.201, *N* = 132, *p* < 0.021) and ACE-R (ϱ = 0.273, *N* = 132, *p* < 0.002) scores and a weak positive correlation was found.

Furthermore, the results showed that there was no statistically significant correlation in individual groups of respondents (Table 3).

### 3.3. Vitamin D and Predictor Variables

All variables of importance for the study (sociodemographic characteristics, laboratory parameters, MoCa, and ACE-R tests) were included in a binary logistic regression model, in order to determine possible predictive variables associated with vitamin D. The results showed that only two covariates are associated with vitamin D: GGT (*OR* 0.38, 95% CI = 0.14–0.99, *p* < 0.05) and ACE-R scale (*OR* 1.09, 95% CI = 0.96–1.20, *p* < 0.05) (Figure 3a,b). R² = 13.2% of the variance in vitamin D levels is explained by our model.

## 4. Discussion

Our findings showed a weak correlation between serum vitamin D levels and cognitive function across all three groups. In their review article, Tardelli et al. (2017) suggest that findings regarding the association between alcohol intake and serum vitamin D levels are heterogeneous and that they equally demonstrate both positive and negative correlations. The latest findings in large studies indicate positive correlations, while earlier population-based studies on specific groups found more negative correlations (association between vitamin D levels in alcohol-dependent individuals and healthy controls) [21]. Respondents in this study exhibited similar sociodemographic characteristics according to the rehabilitation center data: a higher prevalence of men than women, lower education levels, and high unemployment rates [22]. The lower employment rate of the alcohol-dependent respondents can be explained partly by their lifestyle and poor work experience. The majority of respondents in all three groups finished secondary school, their material status was average and they live in urban areas. Most of the alcohol-dependent respondents were middle-aged. Likewise, other studies have shown that vitamin D deficiency is present among the middle-aged population [23,24].

A recent study was conducted to investigate serum vitamin D levels in a sample of 55.844 Europeans and it showed that in 13% of the cases, vitamin D levels were lower than 30 nmol/L [25]. In the present study, all three groups exhibited low vitamin D levels. These findings are similar to other studies, which show that vitamin D deficiency is more present in alcohol-dependent individuals [12,26,27], including patients who suffer from mental disorders in comparison to the general population [10,28]. Furthermore, alcohol-dependent individuals often exhibit low vitamin D levels as a direct result of how alcohol exerts its influence on vitamin D metabolism (malabsorption of vitamin D, decreased levels of the vitamin-D binding protein, reduced ability to hydroxylate vitamin D in the liver), altered biliary excretion, insufficient food intake, or reduced sun exposure. Studies conducted on animals have shown that chronic alcohol use may result in decreased serum concentrations of 1,25-dihydroxyvitamin D due to impaired renal synthesis and/or increased degradation of 1,25-dihydroxyvitamin D. Further research is needed to identify different mechanisms which contribute to low vitamin D levels associated with excessive alcohol use or intoxication [12,27,28,29]. Chun-Qiu Hu et al. (2020) conducted a study on animals and found that vitamin D deficiency exacerbates cellular apoptosis in alcohol-induced liver damage (it leads to hepatic inflammation and oxidative stress—through regulation of oxidative and antioxidant enzymes) [29]. There are also studies showing that excessive alcohol use is associated with increased serum levels of 25 (OH) D [30,31]. It has been suggested that alcohol use can block the conversion of 25-OH D to 1,25-dihydroxyvitamin D [32] and that high levels of 25-OH D are therefore associated with excessive alcohol use [31]. However, these reported findings need to be taken with caution because 1,25-dihydroxyvitamin D is actually the active vitamin D metabolite.

With the discovery of vitamin D receptors and vitamin D hydroxylase in the brain, there has been an increase in the number of studies on the possible association between vitamin D and cognitive function [32]. Data obtained from studies on animals and in vitro indicate that vitamin D plays a role in brain function by regulating neurotrophic factors, neurotransmission, and influencing synaptic plasticity [33,34]. However, there is a lack of consensus among population-based epidemiological studies on the association between serum vitamin D deficiency and cognitive impairment [16,35,36,37]. The reason for this inconsistency may well lie in the fact that it is very difficult to measure an individual’s vitamin D levels in the long term. Serum concentration of 25-OH D is the accepted index of vitamin D status and its life expectancy is approximately two-three weeks, while cognitive impairment is a gradual and long-term process [33]. Cognitive screening is useful when a more detailed neuropsychological assessment cannot be performed or is inadequate. A number of cognitive screening tools are available. These also include ACE-R and MoCa sensitivity tests [38]. Both instruments were validated to assess cognitive impairment associated with substance abuse [19,39,40,41]. It is recommended that cognitive screening be included at the onset of assessment of alcohol-dependent individuals, with further assessment needed for those who exhibit objective impairment [42]. It also indicated that MoCA and ACE-R are valid and time-efficient tools for detecting cognitive impairment in alcohol misuse. Regarding the association between vitamin D levels and cognitive function, there was statistical significance between vitamin D deficiency on overall MoCA and ACE-R scores in all groups. However, separate group results demonstrated no statistical significance between vitamin D levels and cognitive impairment in alcohol-dependent individuals. Although there are practically no literature data on the association between vitamin D levels and cognitive impairment in alcohol-dependent individuals, a lot of research has been completed in this field that demonstrates the association in different population groups. Accordingly, Pettersen (2016) found a correlation between vitamin D levels and verbal fluency in healthy controls [43]. Another study confirmed the association between cognitive impairment and hypovitaminosis D in patients with a major depressive disorder [44]. Some other studies have also found a positive correlation between vitamin D deficiency and cognitive impairment [15,45,46,47]. In another randomized, double-blind, placebo-controlled trial performed on the mental health of a group of 50 adolescents, the effect of vitamin D supplementation on vitamin D status, executive function, and mental health during the winter period was evaluated. The trial demonstrated that respondents with low vitamin D levels exhibited poorer results on cognitive tests. Intake of vitamin D supplements improved their performance on the most demanding tests [48]. Yet, two different clinical trials with samples of over 10,000 subjects failed to prove the association between low vitamin D levels and cognitive impairment. One trial included respondents with a mean of 62 years [49], and the other recruited three different age groups (adolescents, adults, and elderly) [50]. As for this study, to the best of our knowledge, no research to date has examined the association between vitamin D deficiency and cognitive impairment in the alcohol-dependent population.

Furthermore, our findings show that GGT proved to be a significant predictive factor in vitamin D status. In other words, the higher GGT, the lower the vitamin D levels. Some studies have confirmed that reduced serum vitamin D levels are associated with ethanol-induced liver damage, which was manifested by abnormal serum AST values, steatosis, and liver cirrhosis. However, the specific role of vitamin D in pathogenesis, disease progression, and complications during alcoholic liver disease is not yet fully understood [51,52]. A cross-sectional study involving a sample of 24,229 American adults examined the association between vitamin D levels and four liver enzymes (AST, ALT, GGT, and AP-alkaline phosphatase). The results confirmed that lower levels of vitamin D were associated with the presence of higher abnormal values of liver enzymes, mainly AP, but not with the remaining three enzymes. In addition, the correlation was much stronger in men, obese people, people who consume alcohol, and people suffering from viral hepatitis [53]. Another study investigating the association between vitamin D deficiency and liver enzymes (ALT, GGT, AP) in subjects without liver disease did not confirm the correlation between these parameters [54]. Given the current lack of research on the association between vitamin D and liver enzymes, especially GGT in alcohol-related disorders, there is a need for further studies to elucidate the precise mechanisms of the role of vitamin D in alcohol-induced hepatotoxicity. In this study the ACE-R test battery stood out as a predictive factor of vitamin D levels, i.e., the higher the score on the scale, the higher the vitamin D levels. One study by a group of Indian authors confirmed a positive correlation between vitamin D status and scores on the MMSE and ACE-R screening tools in patients with mild cognitive impairment and dementia [55]. Another retrospective cohort study examined the influence of vitamin D status on the recovery of subjects after traumatic head injuries. The results confirmed that subjects with vitamin D deficiency had significantly lower ACE-R scores compared to those exhibiting insufficient or sufficient vitamin D levels. Vitamin D concentrations were also significantly correlated with performance on the ACE-R test battery [56]. On the other hand, there is a lack of research on the association between vitamin D status and ACE-R scores in alcoholics, which indicates the need for further research.

Notwithstanding, there are several limitations to the study. First, it included only a small number of respondents, in particular those with cognitive impairment. Respondents with cognitive impairment were slightly older than the other two groups. Secondly, only a small sample of female respondents was recruited. A low vitamin D level detected in the control group of respondents is also a confounding factor. Since this is a case-control study, the cause-and-effect relationship between hypovitaminosis D and the cognitive function of alcohol-dependent individuals needs to be further investigated in prospective studies. Screening tools for assessing cognition can have false-positive and false-negative results. An additional limitation was the measurement method of vitamin D. In direct comparisons, immunoassay methods show bias and increased variability relative to a liquid chromatography-tandem mass spectrometry (LC-MS/MS), which was recognized as the gold standard. In addition, the present study was carried out in the period from October to March when the northern hemisphere has less exposure to sunlight and as a result, the expected vitamin D levels were lower. Furthermore, other confounding factors are to be considered too: skin type, body mass index, liver, and renal function, geographical location of residence, vitamin D intake, physical activity, smoking, and use of sunscreen. Some of these parameters were taken into consideration but were not mentioned due to the limitations of presenting the overall results.

## 5. Conclusions

In summary, the present study demonstrated vitamin D deficiency across all study groups and at the same time, overall vitamin D deficiency in alcohol-dependent individuals. There were positive correlations between vitamin D deficiency and cognitive function, but with no statistical significance in the alcohol-dependent groups. Due to the limited data in this field, the study provides insight into the potential role of vitamin D and its association with cognitive impairment in alcohol-dependent individuals. As such, it gives rise to future research in this field.

## Figures and Tables

**Figure 1 healthcare-10-01772-f001:**
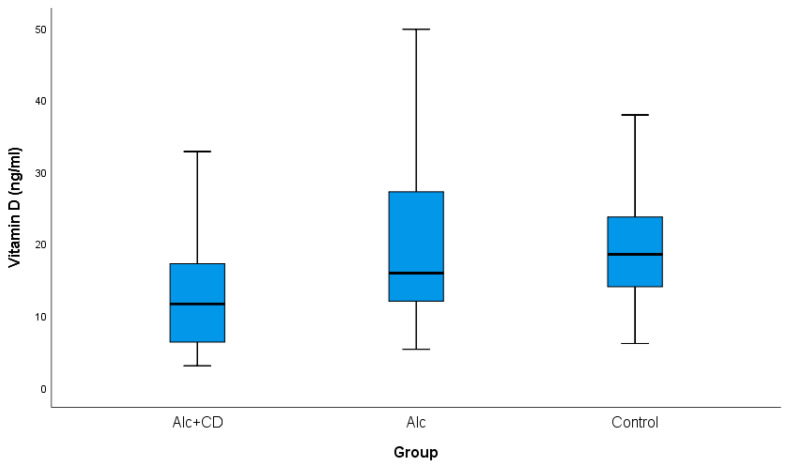
Vitamin D levels by groups. **Notes:** Alc + CD—alcohol-dependent individuals with cognitive deficiency; Alc—alcohol-dependent individuals without cognitive deficiency; Controls.

**Figure 2 healthcare-10-01772-f002:**
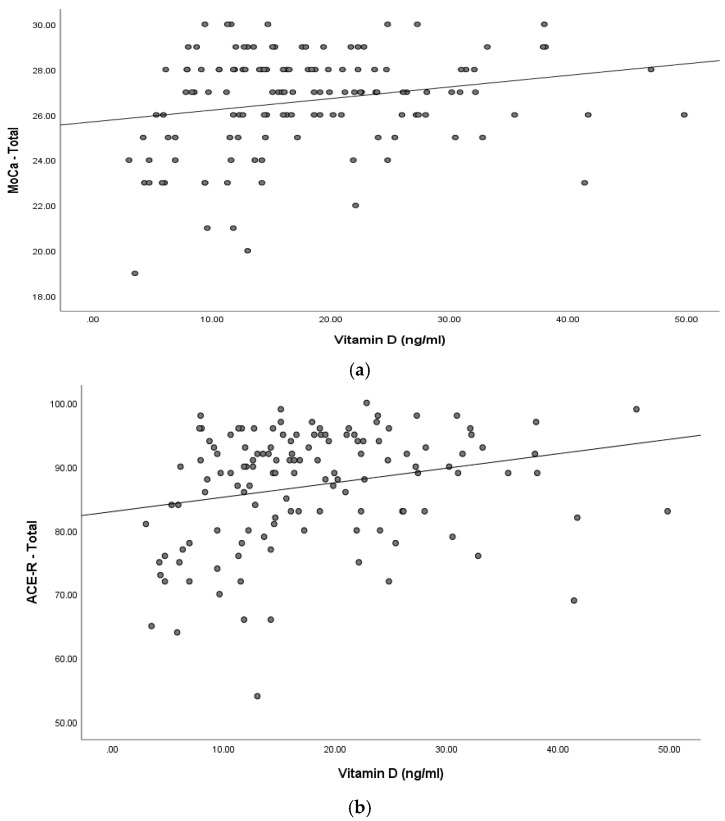
(**a**) Correlation between serum vitamin D levels and cognitive function by groups on the MoCa test. **Note:** MoCa Total- the Montreal Cognition Assessment total score. (**b**) Correlation between serum vitamin D levels and cognitive function by groups on the ACE-R scale. **Note:** ACE-R Total—Addenbrooke’s Cognitive Examination—Revised total score.

**Figure 3 healthcare-10-01772-f003:**
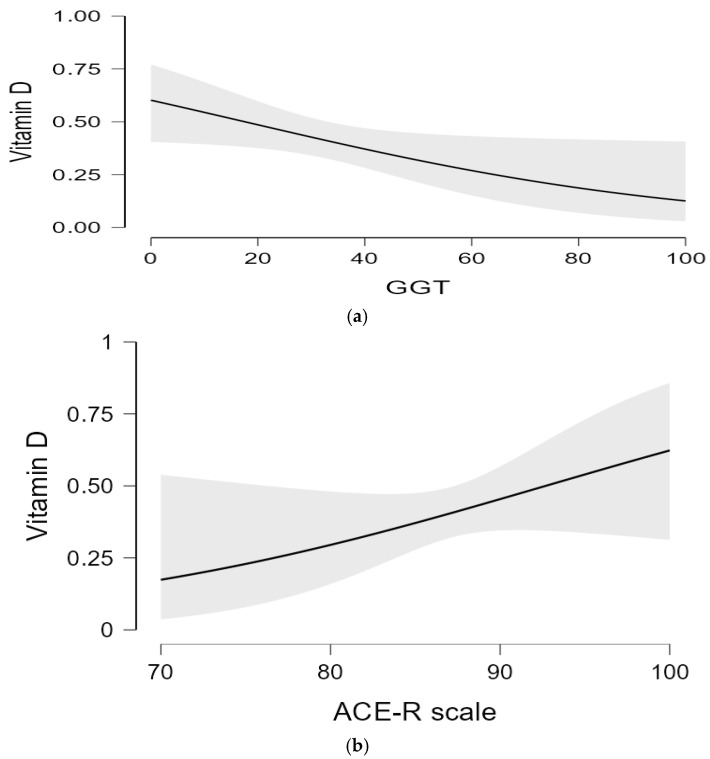
(**a**) Regression analysis of gamma glutamyltransferase. **Note**: GGT—gamma glutamyltransferase (**b**) Regression analysis of ACE-R scale. **Note:** ACE-R Total—Addenbrooke’s Cognitive Examination—Revised total score.

**Table 1 healthcare-10-01772-t001:** Sociodemographic characteristics of study population.

Profile	Number (%) of Respondents			Total	*p*
	Alc+CD	Alc	Control	(*N* = 132)	
**Gender**					*p* = 0.909
Male	26 (78.8)	26 (78.8)	54 (81.8)	106 (80.3)	
Female	7 (21.2)	7 (21.2)	12 (18.2)	26 (19.7)	
**Mean age ± SD**	53.64 ± 3.9	50.73 ± 7.7	50.47 ± 4.8	51.33 ± 5.6	*p* = 0.023
**Marital status**					*p* = 0.014
Married	13 (9.8)	16 (12.1)	44 (33.3)	73 (55.3)	
Divorced	5 (3.8)	10 (7.6)	4 (3.0)	19 (14.4)	
Single	13 (9.8)	6 (4.5)	16 (12.1)	35 (26.5)	
Widow/widower	2 (1.5)	1 (0.8)	2 (1.5)	5 (3.8)	
**Education**					*p* = 0.000
Elementary	14 (10.6)	6 (4.5)	1 (0.8)	21 (15.9)	
Secondary	17 (12.9)	24 (18.2)	59 (44.7)	100 (75.8)	
Higher	2 (1.5)	3 (2.3)	6 (4.5)	11 (8.3)	
**Employment**					*p* = 0.000
Unemployed	12 (36.4)	20 (60.6)	5 (7.6)	37 (28.0)	
Employed	12 (36.4)	9 (27.3)	59 (89.4)	80 (60.6)	
Retired	9 (27.3)	4 (12.1)	2 (3.0)	15 (11.4)	
**Material status**					*p* = 0.000
Below average	12 (36.4)	8 (24.2)	1 (1.5)	21 (15.9)	
Average	20 (60.6)	23 (69.7)	54 (81.8)	97 (73.5)	
Above average	1 (3.0)	2 (6.1)	11 (16.7)	14 (10.6)	
**Place of residence**					*p* = 0.058
Rural	12 (9.1)	8 (6.1)	10 (7.6)	30 (22.7)	
Urban	21 (15.9)	25 (18.9)	56 (42.4)	102 (77.3)	

**Notes:** Alc + CD—alcohol-dependent individuals with cognitive deficiency; Alc—alcohol-dependent individuals without cognitive deficiency; Control.

**Table 2 healthcare-10-01772-t002:** Laboratory parameters of the respondents.

Laboratory Parameters with Reference Range	Alc + CD	Alc	Control	*F*	*p*
	Mean ± SD	Mean ± SD	Mean ± SD		
Blood sedimentation (0–12 mm/h)	9.2 ± 6.1	8.8 ± 9.5	7.7 ± 3.6	1.999	0.642
Red Blood Count (3.86–5.08 × 10¹²/L)	4.4 ± 0.5	4.7 ± 0.5	5.3 ± 0.4	27.148	0.000
White Cell Count (3.40–9.70 × 10^9^/L)	7.1 ± 1.4	7.6 ± 1.6	8.2 ± 1.9	10.709	0.016
Hemoglobin (119–157 g/L)	142.4 ± 12.2	146.3 ± 9.5	152.7 ± 9.1	9.958	0.000
Hematocrit (0.36–0.47 L/L)	0.4 ± 0.04	0.4 ± 0.03	0.4 ± 0.04	8.797	0.000
Platelet Count (158–424 × 10^9^/L)	213.6 ± 58.7	238.3 ± 67.09	255.7 ± 47.16	6.852	0.001
Urea (2.5–6.7 mmol/L)	4.4 ± 1.6	4.5 ± 1.6	5.2 ± 1.4	4.897	0.100
Creatinine (44–88 μmol/L)	69.8 ± 10.8	71.9 ± 11.9	88.0 ± 12.2	38.912	0.000
Glucose (3.9–5.9 mmol/L)	5.1 ± 0.6	4.9 ± 0.7	4.7 ± 0.6	6.604	0.148
Aspartate aminotransferase (0–35 U/L)	31.2 ± 11.6	29.0 ± 9.3	17.9 ± 5.3	46.715	0.000
Alanine aminotransferase (0–45 U/L)	34.6 ± 15.3	31.9 ± 11.4	19.9 ± 9.0	32.472	0.000
Gamma glutamyltransferase (2–55 U/L)	52.4 ± 23.6	42.4 ± 24.4	22.3 ± 9.3	39.836	0.000
Potassium (3.5–5.1 mmol/L)	4.6 ± 0.5	4.6 ± 0.3	4.3 ± 0.4	7.608	0.042
Calcium (2.02–2.55 mmol/L)	2.3 ± 0.1	2.3 ± 0.1	2.3 ± 0.1	1.102	0.235
Magnesium (0.65–1.05 mmol/L)	0.8 ± 0.1	0.8 ± 0.1	0.9 ± 0.1	35.600	0.000
Sodium (136–146 mmol/L)	140.5 ± 2.5	140.3 ± 2.0	140.4 ± 2.6	0.343	0.953
Chloride (98–107 mmol/L)	100.9 ± 3.9	100.6 ± 2.9	99.9 ± 2.7	1.815	0.426
Phosphorus (0.84–1.45 mmol/L)	1.09 ± 0.1	1.1 ± 0.2	1.1 ± 0.2	0.073	0.951

**Notes:** mean (SD); mm/h—millimeters in one hour; mmol-millimole; μmol—micromole; g—gram, L—liter, U—unit; Alc + CD—alcohol-dependent individuals with cognitive deficiency; Alc—alcohol-dependent individuals without cognitive deficiency; Controls.

**Table 3 healthcare-10-01772-t003:** Correlations between vitamin D levels and tests by individual groups.

25-OH Vitamin D		MoCa—Total	ACE-R—Total
**Alc + CD**	* **ƍ** *	0.268	0.201
	* **p** *	0.131	0.263
	* **N** *	33	33
**Alc**	* **ƍ** *	−0.092	−0.050
	* **p** *	0.61	0.782
	* **N** *	33	33
**Control**	* **ƍ** *	−0.144	0.015
	* **p** *	0.247	0.903
	* **N** *	66	66

**Notes:** *ƍ*—Spearman’s rank Correlation; Alc + CD—alcohol-dependent individuals with cognitive deficiency; Alc—alcohol-dependent individuals without cognitive deficiency; Controls; MoCa—the Montreal Cognition Assessment; ACE-R—Addenbrooke’s Cognitive Examination—Revised.

## Data Availability

The datasets used and analyzed during this study are available from the corresponding author upon reasonable request. The data were not publicly available because of ethical considerations.

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
