# Peer review of "Association between Vitamin D and Cognitive Deficiency in Alcohol Dependence"

_healthcare, 2022, doi:10.3390/healthcare10091772_

Round 1

Reviewer 1 Report

This manuscript by Baljak et al reports a case control study assessing the relationship between vitamin D content and cognitive impairment in alcohol dependence. The authors found lower levels of vitamin D in alcohol dependent subjects with cognitive impairment, and an overall correlation between vitamin D content and cognitive function across all groups. The authors explain that this is the first report linking vitamin D levels and cognitive function in alcohol dependent subjects.

Strengths:

This study appears to be designed and performed well. The study design, especially the inclusion and exclusion criteria are thorough and logical. The results are presented in a very clear manner.

Limitations:

(1) Several demographic factors (e.g., level of education and employment) differed between groups. The authors acknowledge these as confounding factors, but I am wondering if it is possible to differentiate the differences in vitamin D levels as being related to alcohol intake versus education/employment level. Or are all three factors interrelated mechanistically?

(2) Minor editing for English spelling and grammar is required. I mainly noticed this in the Abstract.

(3) The sentence on line 122 about respondents less than 12 years of age does not seem to belong in this manuscript. I am not sure if any respondents were < 12 years old (I hope not), and I’m not sure how the 1 point awarded is useful in this cognitive test in this study.

(4) The last paragraph of the Discussion section seems too long. I suggest having the section on study limitations on line 304 be the beginning of a new paragraph.

Author Response

Dear reviewer,

thank you for all your inputs. Please see the attachment.

Kind regards,

Visnja Banjac Baljak

Reviewer 2 Report

Dear Authors
the paper addressed an important and potentially relevant topic.
Only one comment: the final results should be addressed by a multivariate logistic analysis

Minor comment:

paper that reported pleiotropic effects of vitamin D could be emphasized,
for example

Vitamin D in older population: new roles for this 'classic actor'?
Lauretani F, Maggio M, Valenti G, Dall'Aglio E, Ceda GP.Aging Male. 2010 Dec;13(4):215-32

Author Response

Dear reviewer,

thank you on your inputs. Please see the attachment.

Kind regards,

Visnja Banjac Baljak

Round 2

Reviewer 2 Report

No further comments